# Research on the Coordination of Quality Behavior of Supply 3 Chain of E-Commerce Platform under C2B Model of High-Grade E-Commerce Based on Differential Game

**Bin Xu, Zhouhao Zhang \* and Xinqi Li**

School of Management, Shanghai University, Shanghai 200444, China
* Correspondence: zhangzh981017@163.com

**Abstract:** With the increasing demands of consumers for product grade, the C2B model of high-grade e-commerce emerges as required. In order to solve the problem of coordination and cooperation between e-commerce platforms and manufacturers, and to further develop the C2B model of high-grade e-commerce, this paper studies the coordination of supply chain interests by establishing a differential game model considering product grade factors. By comparing the equilibrium solutions of differential games under different decision-making situations, a cooperative coordination mechanism is proposed. Next, the equilibrium solution is further analyzed by means of numerical simulation. Finally, the influence of several important parameters on the equilibrium solution is discussed through sensitivity analysis. It is found that (1) the supply chain parties have optimal quality behavior in the centralized decision-making situation, and the overall benefit is the greatest. (2) Compared with the Nash non-cooperative game, the optimal quality behavior of the dominant party remains unchanged, and the optimal quality behavior of the following party is enhanced after both parties move from the Stackelberg master-slave game, and the optimal profits of both parties, as well as the overall increase. (3) The cooperative coordination model can coordinate the quality behavior of both sides of the supply chain when the benefit distribution coefficient is within a specific range.

**Keywords:** E-commerce; C2B model of high-grade E-commerce; differential game; quality behavior; coordination mechanism





## 1. Introduction

With the rapid spread of the Internet, commerce can circulate in a broader field. E-commerce integrates traditional offline business resources into a new mode and sells products to consumers worldwide [1]. In recent years, consumers have had increasingly high requirements for the product's appearance, quality, functionality, and other product-grade (PG) aspects. The original promotion of high-grade products may be reduced to low-grade products that no one bought in a short time. If not updated in time, they will be eliminated by the market. Therefore, to meet the consumers' pursuit of PG, the core of the E-commerce platform (EP) operation is gradually shifted from the enterprise website to the customer, and customer demand has also become a core point of enterprises needing more concerns [2], i.e., consumer-centric, consumer-driven, value-collaborative quality E-commerce C2B(Customer-to-Business) model. This is an emerging E-commerce model, i.e., the customer-to-maker E-commerce model [3]. The C2B model is first triggered by consumers' consumption demand, followed by manufacturing enterprises organizing production activities according to consumers' consumption demand, changing the traditional E-commerce market to a production-led consumption concept [4]. Manufacturing companies also urgently need to apply various emerging technologies to continuously optimize and update products to improve the responsiveness of customer needs and adapt to the complex market competition environment [5]. Currently, many EPs have launched high-grade E-commerce brands with reference to the C2B model for people's

daily needs, such as Taobao Xinxue, Xiaomi Youpin, Jingtongzuo, and Netease Yanxue. The aforementioned brands can stand on the market attribute to the following three reasons: (1) the assurance of PG, each product of these E-commerce brands has been carefully manufactured and selected; (2) the guarantee of cost-effectiveness, these E-commerce brands have a clear commodity positioning, serving consumers who have a certain pursuit of PG, thereby gaining a large number of loyal customers; (3) the concern of critical sensibility, EPs will also pay attention to the appearance design when manufacturing products and make them into arts. EPs use big data, cloud computing, and other technologies to explore user needs, develop products with precise design, and then cooperate with global high-level manufacturing enterprises to optimize production costs and ensure product quality. As a result, PG is constantly improved through frequent negotiations between EPs and manufacturers. However, this form of cooperation and negotiation inevitably leads to a conflict of interest and the allocation of interests between the two sides [6]. In the trade cooperation between EPs and manufacturers of various commodities, frequent negotiation undoubtedly increases the workload and costs much time for EPs. Based on the above business model, we study how the EP and its manufacturers under the C2B model implement cooperation to maximize the interests of both parties. However, only a reasonable distribution can ensure their maximum benefits, which is also a critical difficulty in negotiating during cooperation. Therefore, this paper explores how to control the cost of the commodity produced by manufacturers to minimize their costs of products to meet the PG requirements of EPs for their E-commerce brands. We study the coordination of quality behavior (QB) of the secondary supply chain (SSC) composed of EPs and their manufacturers. The research objective of this paper is to investigate a method of increasing the profits of supply chain (SC) members, thereby improving the efficiency and stability of cooperation between E-commerce platforms and manufacturers. This research provides contribution and reference value to the development of the E-commerce industry.

Based on the above background, this paper is based on the SSC composed of an EP and its cooperative manufacturers and integrates the factors of PG. This paper investigates how both parties make decisions to reasonably distribute benefits while maintaining stable cooperation and constructs differential game (DG) models under centralized and decentralized decision-making, respectively, for comparative analysis. We propose a better coordination mechanism for cooperation between EPs and their manufacturers.

The theoretical implication of this paper includes the following aspects: First, this paper proposes a DG model of the SSC composed of an EP and its cooperative manufacturers. With PG innovation as an important influencing factor, we construct correlation variables to reflect the dynamic changes in product quality over time. Secondly, this paper investigates the equilibrium solutions of the model under centralized and decentralized decision-making through DGs. In different situations, both parties may make decisions as an ensemble or as individuals and then proposes a more realistic coordination mechanism for the cooperation between EPs and their manufacturers after comparative analysis. Thirdly, this paper conducts sensitivity analysis on three critical parameters, namely QB impact coefficient, quality cost coefficient, and PG impact coefficient on revenue, and the equilibrium solution of the DG under the cooperative coordination mechanism. Thus, we can grasp the relevant laws to increase the total revenues of the SC.

In addition, this paper includes the following practical implications: First, this paper analyzes how EPs and manufacturers cooperate under decision-making scenarios to maximize benefits through equitable cost sharing. It has implications for the cooperation model of SC members under the C2B model of quality E-commerce. Secondly, the paper establishes a cooperation and coordination mechanism between the EP and the manufacturer to maximize the overall benefits of the whole SC under centralized decision-making situations. This cooperative coordination mechanism can effectively motivate the members to participate spontaneously in the collaboration and coordination mode and also helps to enhance the stability of the SC. Third, this paper introduces a series of comparative analyses. In the SSC network built by the EP and its partner manufacturers, we propose feasible

operational strategies to the relevant enterprises, which are of great reference significance to business operators.

The remainder of this paper is organized as follows: Section 2 of the paper conducts a review of relevant literature; Section 3 describes the problem and proposes model hypotheses; Section 4 constructs DG models of QB in centralized and decentralized decision situations, respectively; Section 5 conducts a comparative analysis of centralized and decentralized decision situations and establishes a cooperative coordination mechanism between EPs and their manufacturers; Section 6 validates the results through numerical simulations and conducts parameter sensitivity analysis; Section 7 summarizes the findings of the paper and gives corresponding suggestions and insights.

## 2. Literature Review

This paper mainly refers to three types of literature, and one is the study of EPs and the C2B model. In the past decade, the stable EP, with Amazon as the generator, began to develop rapidly, and retailers can choose to sell their products on their official websites or with the help of EPs such as Amazon. Retailers and EP companies must make optimal decisions to ensure maximum benefits [7]. The emergence of E-commerce has enriched the choice of shopping channels for consumers. Its development has prompted companies to accelerate the integration of online and offline channels, such as Dell, Corum, and Nike, and other manufacturing companies have not only established their own online direct sales channels but also cooperated with major EPs so that consumers can not only experience consumption in offline channels but also purchase the required products in online channels without leaving home [8]. In the SC consisting of manufacturers and EPs, many factors can influence them, such as third-party, public information such as product reviews can influence the sales patterns of Eps [9], and the marketing ability of sellers can also have an impact on consumer behavior [10]. In addition, EPs may also select or develop different business models according to the current market conditions. EPs allow manufacturers to transact directly with customers and charge corresponding fees; a form often referred to as the agency sales model [11]. Depending on the degree of product competition, one may choose a pure reselling model, a pure reselling model, or a hybrid model [12]. Some scholars have studied the influence of SC power structure on manufacturers' impact on optimal distribution channel strategy; the study showed that manufacturers prefer retailers to follow a bulk ordering strategy. When retailers do not have market dominance, the direct sales model is more profitable for both parties [13]; the choice of an online sales model also depends on whether the EP or the supplier can collect information about the product marketing activities [14]. Some EPs will adopt a trade-in strategy to improve their revenue [15], and EPs that only sell their products will consider opening their platform to third-party sellers [16]. With the development of EPs, consumers' demand for personalized, entertaining, and scenario-based products is increasing, and various C2B emerging E-commerce models oriented to consumers' personalized needs are gradually emerging [17]. Some scholars have developed a group-buying site dependency model by integrating the EP success model and commitment-trust theory to study the critical determinants of SC dependency under the C2B model [18]. Coordination mechanisms can also be established in the field of new EPs under the C2B model to improve the overall SC profitability [19].

Another category of literature is the study of quality issues in the SC. Product quality is the basis for voluntary consumer payment and an essential basis for product pricing by manufacturers. Quality is not only the basis for business survival and growth but also a critical issue that cannot be ignored in SC management decisions [20]. The focus of quality management is also gradually shifting to the SC [21]. The impact of supplier decision sequencing regarding quality investment and whole-sale price in a two-level SC with uncertain demand has been discussed by some scholars, and the optimal supplier decision was found to depend on the cost structure [22]. In a three-level SC consisting of suppliers, manufacturers, and retailers, the degree of quality ambiguity and advertising sensitivity can impact the optimal profit allocation among agents [23]. Trust and information sharing

among SC firms have also been shown to assess important indicators of SC quality performance [24]. Supply chains involve steep competition between retailers and manufacturers and horizontal competition between manufacturers [25]. Therefore, a collaborative SC mechanism can be established to use the presence of quality management as a modifiable mediating effect that allows SC firms to have more significant opportunities to grow in the face of intense competition [26]. Incentives for collaborative quality improvement can also be established in buyer-driven SCs to improve the overall performance of the SC [27]. In addition, some scholars have collaborated with several companies to obtain their operational data and found that quality management practices affect the growth of SC benefits based on the patterns presented by the data [28]. In SSC networks, quality control affects the related alignment of interests [29]. Some studies have designed a collaborative approach to quality prevention and control in a second-tier SC using quality practices, supplier linkages, and leadership strength as key variables [30].

The last category of the literature summarizes the application of the game theory approach in the study of EPs. Facing the credit problem of EPs, some scholars have discussed the formation of credit risk in EPs based on the game theory approach and made some suggestions for the future credit problem of Eps [31]. In addition, in the logistics market of EPs, when two logistics service providers compete for the number of orders and service level decisions for a specific product, it can be demonstrated that the demand for logistics services depends on different competitive structures by building Cournot game and Stackelberg game models [32]. The forms and types of goods on EPs tend to diversify, and with the widespread interest in sustainability, more and more consumers are buying green products on EPs. A study has constructed an evolutionary game model of green product quality regulation involving government regulators, third-party EPs, online sellers, and consumers [33]. In the fresh produce EP, a three-stage pricing model with third-party cold chain logistics companies as leaders, fresh produce E-commerce companies as followers, and consumers as secondary followers was established based on the Stackelberg game model and interest equilibrium analysis, and the optimal pricing strategy was derived from the study and analysis [34]. Other studies have used game theory models to analyze whether service providers with asymmetric capabilities in the new E-commerce model under O2O should contract with intermediaries to introduce opaque distribution channels and use pricing mechanisms to sell opaque services [35]. In addition, the partner selection of cross-border E-commerce firms under the B2B model can be analyzed through an asymmetric evolutionary game [36]. However, if some aspects of the game process change over time, a DG model is needed to reflect these variables' time-varying characteristics. In this regard, it has been considered to establish a Stackelberg DG model to study the optimal strategy of promotion between EPs and suppliers in the case of product reputation, and the study showed that EPs should strengthen their logistics services, increase their promotion efforts, and try to attract the participation of reputable suppliers, thus leading them to choose the Open Platform Program [37].

By combing through the above three categories of related literature, it is found that most of the research on EPs has explored various factors that affect a specific link or element in the platform SC and how EPs choose the appropriate operating model under different conditions. For the various C2B model EPs that are currently emerging, some scholars have researched the areas of group buying E-commerce and fresh produce E-commerce. However, in general, there is less research on C2B EPs and the exploration of this area of quality E-commerce. Research on SC quality issues has mainly focused on exploring how quality management acts on the SC and what measures should be taken to enhance SC benefits according to the characteristics of the problem. Few studies have extended the concept of quality problems to the QB and studied its impact on the SC. Additionally, in the research area of EPs, most scholars have constructed the DG model to study the decision-making problems of multiple parties in the SC under different business models. Some researchers have also investigated DG models in certain EP areas, which feature the introduction of variables with dynamic changes over time. However, fewer research

elements introduce DG models with variables that change dynamically over time. Overall, few studies consider PG and analyze the QB of SC members by building DG models in the C2B model of high-grade E-commerce. This paper largely complements this theoretical gap.

### 3. Problem Description and Model Assumptions

Consider a secondary single-channel SC consisting of an EP and a manufacturer for a continuous period $t \in [0, \infty)$. The high-grade E-commerce C2B model first analyzes user needs and preferences through the platform's big data. Based on the insights into consumer needs, the EP team will redesign the product. An aspect of the product characteristics is adjusted or optimized, and the product design requirements are sent to the manufacturer. The manufacturer then purchases the raw materials required to produce the product and proceeds with production activities and inspection. After passing the inspection, the product is sent to the EP, which is finally responsible for checking the PG's goodness and acceptance. It can be seen that the grade of the products purchased by the consumers depends on the QB of both the EP and the manufacturer. The following assumptions are made based on the above analysis and modeling needs.

(1) Both EPs and manufacturers are rational individuals pursuing their interest maximization and risk-neutral responses to risks. Both parties know each other's cost and profit information, and the game process of both parties is a perfect and perfect information game.

(2) In this paper, PG refers to product quality, appearance, and function factors that can increase consumers' desire to buy the goods. The QB refers to the internal behavior of the company that involves changing PG, such as designing the product's appearance, adjusting the product's production line, etc. Enhancing the QB or increasing the QB means that the operation has a positive effect on PG, and PG increases; weakening the QB or decreasing the QB means that the operation hurts PG, and PG decreases.

(3) On the one hand, as consumers' desire for a high quality of life increases, EPs and their manufacturers will enhance their QB, thus increasing PG; on the other hand, with the rapid development of the times and technological progress, the grade of products and consumers' vision in examining products will also continue to improve. The quality products in the past may also be gradually called ordinary grade products, which can be considered as the decline of PG over time; meanwhile, EPs and manufacturers will constantly update their products in cooperation. Therefore, PG changes dynamically with time, and the following differential equation can express its change pattern.

$$\dot{Q}(t) = \alpha x(t) + \beta y(t) - \gamma Q(t)$$

where $Q(t)$ denotes the PG at moment $t$, and $Q(0) = Q_0$ is the PG at the initial moment; $x(t)$ and $y(t)$ denote the QBs of the manufacturer and the EP, respectively; $\alpha$ and $\beta$ denote the influence coefficients of the QBs of the manufacturer and the EP on the PG, respectively, and their magnitudes indicate the strength of the influence of the QBs of the manufacturer and the EP and the size of the improvement of the PG; $\gamma > 0$ denotes the decline rate of the PG over time.

(4) QB enhancement of a firm costs money, especially the QB of a manufacturer, so quality cost is used to represent the cost to enhance a specific QB. It is assumed that the cost functions of the EP and its manufacturer have convex characteristics, i.e., the cost of QB is proportional to the QB. Therefore, the quality cost functions of the manufacturer and the EP can be expressed as follows, respectively

$$c(x) = \frac{1}{2}\lambda x^2(t)$$

$$c(y) = \frac{1}{2}\mu y^2(t)$$

Moreover, the equations satisfy $\frac{dc(x)}{dx} > 0$, $\frac{d^2c(x)}{dx^2} > 0$, $\frac{dc(y)}{dy} > 0$, and $\frac{d^2c(y)}{dy^2} > 0$. where $\lambda$ and $\mu > 0$ denote the quality cost coefficients of the manufacturer and the EP. The quality cost coefficients indicate the magnitude of the quality costs of different levels of impact spent by the manufacturer and the EP to improve the same level of QB. The coefficients affect the quality cost of both parties for improving PG.

(5) According to the hypothesis that "product quality is a linear function of SC revenue", based on the reasonable reality discussed by Xionghui Zhang et al. [38], it can be assumed that the revenue obtained from product SC sales is a linear function of PG, which can be expressed as follows

$$\pi(Q(t)) = r + kQ(t)$$

where $r$ and $k$ are constants >0, $k$ indicates the degree of impact of PG on earnings, and its value is determined by the degree of importance people attach to PG in the general market environment, and the greater the importance of PG, the greater its $k$ value and the more significant its impact on earnings.

(6) Assuming that the discount rate $\rho > 0$ is the same for the EP and the manufacturer, both parties allocate the total SC benefits according to a specific ratio, with the manufacturer's share being $\theta \in (0, 1)$ and the EP's share being $1 - \theta$.

(7) Since the above modeling parameters are time-independent constants, the time $t$ will be omitted below for computational writing convenience.

## 4. QB Game Model Construction

In this section, the DG model of QB is constructed from two aspects, which are the centralized decision-making case and the decentralized decision-making case. Among them, the centralized decision situation is where the EP and the manufacturer act as a decision-making whole. Both parties aim at maximizing the SC benefits, which determines the level of QB of both parties. In the decentralized decision situation, the EP and its manufacturer are two rational and independent individuals who pursue their interests as their goals. The following is the process and analysis of the equilibrium solution.

### 4.1. Centralized Decision-Making Situations

In the centralized decision situation, since the EP and its manufacturer are an organic whole, the objective is to find the optimal level of QB of the EP and its manufacturer in continuous time so that the discounted value of its total benefit is maximized, and the objective function is

$$\int_0^\infty e^{-\rho t}[r + kQ - \frac{1}{2}\lambda x^2 - \frac{1}{2}\mu y^2]dt \tag{1}$$

The optimal value function of Equation (1) satisfies the HJB (Hamilton–Jacobi–Bellman) equation [39], as follows

$$\rho S = \max_{x,y}\{r + kQ - \frac{1}{2}\lambda x^2 - \frac{1}{2}\mu y^2 + S'(\alpha x + \beta y - \gamma Q)\} \tag{2}$$

According to Equation (2) for obtaining the first-order partial derivative of it under the maximum value is obtained as

$$x = \frac{S'\alpha}{\lambda}, y = \frac{S'\beta}{\mu} \tag{3}$$

Substituting Equation (3) into Equation (2), we obtain

$$\rho S = r + kQ - \frac{(S'\alpha)^2}{2\lambda} - \frac{(S'\beta)^2}{2\mu} + S'(\frac{S'\alpha^2}{\lambda} + \frac{S'\beta^2}{\mu} - \gamma Q)$$

After rectification, it is obtained as follows

$$\rho S = r + \frac{(S'\alpha)^2}{2\lambda} + \frac{(S'\beta)^2}{2\mu} + (k - \gamma S')Q \tag{4}$$

From Equation (4), it can be seen that the linear function of HJB is the solution of the equation [39], so that

$$S = a_1 + b_1 Q \tag{5}$$

where $a_1$ and $b_1$ are the coefficients to be determined. Substituting the derivative of Equation (5) concerning into Equation (4) and concatenating it with Equation (5), we obtain

$$a_1 = \frac{1}{\rho}(r + \frac{(k\alpha)^2}{2\lambda(\rho + \gamma)^2} + \frac{(k\beta)^2}{2\mu(\rho + \gamma)^2}), b_1 = \frac{k}{\rho + \gamma} \tag{6}$$

Substituting $S' = b_1$ obtained in Equation (6) into Equation (3), the optimal QB of the manufacturer and the EP can be obtained as

$$x^* = \frac{k\alpha}{(\rho + \gamma)\lambda}, y^* = \frac{k\beta}{(\rho + \gamma)\mu} \tag{7}$$

Further, the optimal value of the product SC in Nash equilibrium can be obtained as

$$S^* = \frac{1}{\rho}(r + \frac{(k\alpha)^2}{2\lambda(\rho + \gamma)^2} + \frac{(k\beta)^2}{2\mu(\rho + \gamma)^2}) + \frac{k}{\rho + \gamma}Q \tag{8}$$

where the SC benefits reach the optimal value, and $Q$ also has a definite function related to time $t$. Substituting the result of Equation (7) into $\dot{Q}(t) = \alpha x(t) + \beta y(t) - \gamma Q(t)$, the particular solution of $Q(0) = Q_0$ can be obtained from the general solution formula of the first-order linear ordinary differential equation as

$$Q^* = \frac{Q_0}{e^{\gamma t}} + (1 - e^{-\gamma t})(\frac{k\alpha^2}{\gamma\lambda(\rho + \gamma)^2} + \frac{k\beta^2}{\gamma\mu(\rho + \gamma)^2}) \tag{9}$$

### 4.2. Decentralized Decision-Making Situations

In the decentralized decision situation, the EP and the manufacturer are two rational and independent individuals who pursue their interest maximization with this premise. In this subsection, we analyze the DG equilibrium solutions for three situations: the Nash non-cooperative game, the manufacturer-dominated Stackelberg master-slave game, and the EP-dominated Stackelberg master-slave game.

- Nash's non-cooperative game model with independent and equal parties

In this case, both parties are independent individuals and equal in status; both are in pursuit of their interest maximization as the goal, and both parties simultaneously choose their best QB level to maximize their interests, then the manufacturer's objective function is

$$\int_0^\infty e^{-\rho t}\{\theta[r + kQ] - \frac{1}{2}\lambda x^2\}dt$$

The objective function of the EP is

$$\int_0^\infty e^{-\rho t}\{(1 - \theta)[r + kQ] - \frac{1}{2}\mu y^2\}dt$$

The calculation process is the same as that in 4.1, which is omitted here.

The optimal level of quantity behavior of manufacturers and EPs under Nash equilibrium is obtained as

$$x_1^* = \frac{\alpha k\theta}{\lambda(\rho + \gamma)}, y_1^* = \frac{\beta k(1 - \theta)}{\mu(\rho + \gamma)} \tag{10}$$

and the optimal PG is

$$Q_1^* = (Q_0 - \frac{\alpha^2 k\theta}{\gamma\lambda(\rho + \gamma)} - \frac{\beta^2 k(1 - \theta)}{\gamma\mu(\rho + \gamma)})e^{-\gamma t} + \frac{\alpha^2 k\theta}{\gamma\lambda(\rho + \gamma)} + \frac{\beta^2 k(1 - \theta)}{\gamma\mu(\rho + \gamma)} \tag{11}$$

The optimal function of the manufacturer, EP, and SC is further obtained as

$$S_{c1}^* = \frac{1}{\rho}\{r\theta + \frac{k^2}{(\rho + \gamma)^2}[\frac{\alpha^2\theta^2}{2\lambda} + \frac{\beta^2\theta(1 - \theta)}{\mu}]\} + \frac{k\theta}{\rho + \gamma}Q \tag{12}$$

$$S_{s1}^* = \frac{1}{\rho}\{r(1 - \theta) + \frac{k^2}{(\rho + \gamma)^2}[\frac{\alpha^2\theta(1 - \theta)}{\lambda} + \frac{\beta^2(1 - \theta)^2}{2\mu}]\} + \frac{k(1 - \theta)}{\rho + \gamma}Q \tag{13}$$

$$S_1^* = \frac{1}{\rho}\{r + \frac{k^2}{2(\rho + \gamma)^2}[\frac{\alpha^2\theta(2 - \theta)}{\lambda} + \frac{\beta^2(1 - \theta^2)}{\mu}]\} + \frac{k}{\rho + \gamma}Q \tag{14}$$

- Manufacturer-driven Stackelberg master-slave game model

Generally, the manufacturer is the core company in the product SC. The QB of the manufacturer will determine the good or bad of PG. Moreover, EPs are more attractive to manufacturers, and it is profitable for manufacturers if they can reach long-term cooperation. Therefore, to stimulate the coordination of QB and achieve the high PG requirement of EPs under the C2B model of high-grade E-commerce, manufacturers will take the initiative to share the cost of QB of EPs to achieve successful cooperation. At this point, it is the Stackelberg master-slave game in which the manufacturer is the leader, and the EP is the follower. First, the manufacturer determines its optimal level of QB and bears the proportion $\delta \in (0, 1)$ of the EP's quality costs, $\delta$ being the quality cost-sharing factor. After observing the manufacturer's response, the EP then determines the QB that maximizes its benefit. Of course, the manufacturer can anticipate the EP's following reaction when determining the QB initially. Therefore, the level of the best QB of the EP is first determined by the reverse induction method.

The objective function of the EP is

$$\int_0^\infty e^{-\rho t}\{(1 - \theta)[r + kQ] - \frac{1}{2}\mu(1 - \delta)y^2\}dt$$

The manufacturer will determine its own best QB based on the best QB of the EP. Its objective function is.

$$\int_0^\infty e^{-\rho t}\{\theta[r + kQ] - \frac{1}{2}\lambda x^2 - \frac{1}{2}k\delta(y^*)^2\}dt$$

The calculation process is the same as that in 4.1, which is omitted here.

The optimal level of QB of manufacturers and EPs in the manufacturer-led Stackelberg game scenario is obtained as

$$x_2^* = \frac{\alpha k\theta}{\lambda(\rho + \gamma)}, y_2^* = \frac{\beta k(1 + \theta)}{2\mu(\rho + \gamma)} \tag{15}$$

$$\delta_1^* = \left\{ \begin{array}{l} \frac{3\theta-1}{1+\theta}, \theta > \frac{1}{3} \\ 0, \theta \leq \frac{1}{3} \end{array} \right. \tag{16}$$

and the optimal PG is

$$Q_2^* = (Q_0 - \frac{\alpha^2 k\theta}{\gamma\lambda(\rho+\gamma)} - \frac{\beta^2 k(1+\theta)}{2\gamma\mu(\rho+\gamma)})e^{-\gamma t} + \frac{\alpha^2 k\theta}{\gamma\lambda(\rho+\gamma)} + \frac{\beta^2 k(1+\theta)}{2\gamma\mu(\rho+\gamma)} \tag{17}$$

The optimal value function of the manufacturer, EP and SC under manufacturer domination is further obtained as

$$S_{c2}^* = \frac{1}{\rho}\{r\theta + \frac{k^2}{2(\rho+\gamma)^2}[\frac{\alpha^2\theta^2}{\lambda} + \frac{\beta^2(1+\theta)^2}{4\mu}]\} + \frac{k\theta}{\rho+\gamma}Q \tag{18}$$

$$S_{s2}^* = \frac{1}{\rho}\{r(1-\theta) + \frac{k^2(1-\theta)}{(\rho+\gamma)^2}[\frac{\alpha^2\theta}{\lambda} + \frac{\beta^2(1+\theta)}{4\mu}]\} + \frac{k(1-\theta)}{\rho+\gamma}Q \tag{19}$$

$$S_2^* = S_{c2}^* + S_{s2}^* = \frac{1}{\rho}\{r + \frac{k^2}{2(\rho+\gamma)^2}[\frac{\alpha^2\theta(2-\theta)}{\lambda} + \frac{\beta^2(3-\theta^2+2\theta)}{4\mu}]\} + \frac{k}{\rho+\gamma}Q \tag{20}$$

- The Stackelberg master-slave game model dominated by EPs

The above is a manufacturer who wants to actively improve QB and share the cost of quality to seek cooperation with EPs. However, there is another situation: the EP as a fledgling E-commerce self-owned brand; although you can rely on the platform's mall to play a certain degree of brand awareness, for some famous manufacturers EP is not necessarily the only partner, not even in their consideration. Therefore, the EP wants to cooperate with these manufacturers to promote cooperation between the two sides and stimulate QB coordination; the EP will choose to share the manufacturer's quality costs. At this point, the EP is the leader; the manufacturer is the follower of the Stackelberg master-slave game. First, the EP determines its optimal level of QB and bears the manufacturer's share of quality costs $\delta \in (0,1)$, where $\delta$ is the quality cost-sharing factor. The manufacturer can observe the EP's decision and adjust its QB to determine the maximum benefit. Of course, EPs can also anticipate the impact of their decisions on manufacturers. Therefore, the manufacturer's optimal level of QB can be determined first based on the inverse induction method.

The manufacturer's objective function is

$$\int_0^\infty e^{-\rho t}\{\theta[r + kQ] - \frac{1}{2}\lambda(1-\delta)x^2\}dt$$

The EP will determine its own best QB based on the manufacturer's best QB. Its objective function is

$$\int_0^\infty e^{-\rho t}\{(1-\theta)[r + kQ] - \frac{1}{2}\mu y^2 - \frac{1}{2}\lambda\delta(x^*)^2\}dt$$

The calculation process is the same as that in 4.1, which is omitted here.

The optimal level of QB of manufacturers and EPs in the case of the EP-led Stackelberg game is obtained as

$$x_3^* = \frac{\alpha k(2-\theta)}{2\lambda(\rho+\gamma)}, y_3^* = \frac{\beta k(1-\theta)}{\mu(\rho+\gamma)} \tag{21}$$

$$\delta_2^* = \begin{cases} \frac{2-3\theta}{2-\theta}, \theta < \frac{2}{3} \\ 0, \theta \geq \frac{2}{3} \end{cases} \tag{22}$$

and the optimal PG is

$$Q_3^* = (Q_0 - \frac{\alpha^2 k(2-\theta)}{2\lambda\gamma(\rho+\gamma)} - \frac{\beta^2 k(1-\theta)}{\mu\gamma(\rho+\gamma)})e^{-\gamma t} + \frac{\alpha^2 k(2-\theta)}{2\lambda\gamma(\rho+\gamma)} - \frac{\beta^2 k(1-\theta)}{\mu\gamma(\rho+\gamma)} \tag{23}$$

The optimal value function of the manufacturer, EP and SC under the domination of EP is further obtained as

$$S_{c3}^* = \frac{1}{\rho}\{\theta r + \frac{k^2\theta}{(\rho+\gamma)^2}[\frac{\alpha^2(2-\theta)}{4\lambda} + \frac{\beta^2(1-\theta)}{\mu}]\} + \frac{k\theta}{\rho+\gamma}Q \tag{24}$$

$$S_{s3}^* = \frac{1}{\rho}\{(1-\theta)r + \frac{k^2}{2(\rho+\gamma)^2}[\frac{\beta^2(1-\theta)^2}{\mu} + \frac{\alpha^2(2-\theta)^2}{4\lambda}]\} + \frac{k(1-\theta)}{\rho+\gamma}Q \tag{25}$$

$$S_3^* = \frac{1}{\rho}\{r + \frac{k^2}{2(\rho+\gamma)^2}[\frac{\alpha^2(4-\theta^2)}{4\lambda} + \frac{\beta^2(1-\theta^2)}{\mu}]\} + \frac{k}{\rho+\gamma}Q \tag{26}$$

## 5. Comparative Analysis and Coordination Mechanism Establishment

In this section, the equilibrium solution of the model under centralized and decentralized decision-making of the DG study obtained in the previous section is compared and analyzed. Then, based on this, a better coordination mechanism for cooperation between both EPs and their manufacturers is proposed to increase the overall benefit further and be more practical.

### 5.1. Contrast Analysis

- Centralized versus decentralized decision-making scenarios

According to the function of the optimal value of the overall SC benefit in four different cases, it is obtained that

$$S^* - S_1^* = \frac{k^2}{2\rho(\rho+\gamma)}[\frac{\alpha^2(1-\theta)^2}{\lambda} + \frac{\beta^2\theta^2}{k}] > 0$$

$$S^* - S_2^* = \frac{k^2(1-\theta)^2}{2\rho(\rho+\gamma)^2}[\frac{\alpha^2}{\lambda} + \frac{\beta^2}{4\mu}] > 0$$

$$S^* - S_3^* = \frac{k^2\alpha^2\theta}{8\lambda\rho(\rho+\gamma)^2} + \frac{k^2\beta^2\theta^2}{2\mu\rho(\rho+\gamma)^2} > 0$$

Corollary 1 can be obtained from the above equation.

**Corollary 1.** *When the profit distribution $\theta \in (\frac{1}{3}, \frac{2}{3})$, the maximum total profit of the product SC can be obtained in the centralized decision-making scenario compared to the three scenarios under decentralized decision-making.*

From Corollary 1, it is clear that SC members no longer focus only on their interests under centralized decision-making. Instead, they realize the information platform and data sharing and interaction in a fully cooperative form and break the communication barriers in the whole SC. To maximize the overall benefit of the SC, the SC members continuously communicate and negotiate the profit distribution between them within a reasonable range, and finally form a solution that satisfies both parties, thus achieving an increase in the overall benefit of the SC and realizing the Pareto optimum. Currently, for secondary and

multi-level SCs, the profit distribution process is distributed throughout the SC. Only by striving to realize centralized decision-making can we build an effective and complete SC benefit-sharing system to better provide consumers with high-grade products and services.

According to the best QB of manufacturers and EPs under four different game scenarios, we can obtain

$$x^* - x_1^* = \frac{\alpha k(1-\theta)}{\lambda(\rho+\gamma)} > 0$$

$$x^* - x_2^* = \frac{\alpha k(1-\theta)}{\lambda(\rho+\gamma)} > 0$$

$$x^* - x_3^* = \frac{\alpha k\theta}{2\lambda(\rho+\gamma)} > 0$$

$$y^* - y_1^* = \frac{\beta k\theta}{\mu(\rho+\gamma)} > 0$$

$$y^* - y_2^* = \frac{\beta k(1-\theta)}{2\mu(\rho+\gamma)} > 0$$

$$y^* - y_3^* = \frac{\beta k\theta}{\mu(\rho+\gamma)} > 0$$

Corollary 2 can be obtained from the above equation.

**Corollary 2.** *When the profit distribution $\theta \in (\frac{1}{3}, \frac{2}{3})$, the best QB of both the manufacturer and the EP under centralized decision is better than the best QB of both parties under the three cases of decentralized decision.*

From Corollary 2, it is clear that with centralized decision-making, SC members will work together to meet the best QB of both parties. In today's market environment, consumers prefer to buy better-quality products and relatively more extended service life. Therefore, in order to occupy the market, SC members can effectively coordinate the QB of each other through various methods, such as in the C2B model of high-grade E-commerce model where EPs and manufacturers jointly set product quality standards. Forming a benign mechanism of joint supervision between the two sides, the QB of all parties is enhanced by the positive boost to promote the further improvement of product quality and meet consumer demand for product quality.

According to the PG function in four different cases, we can obtain

$$Q^* - Q_1^* = \left\{ \frac{k\alpha^2}{\gamma\lambda(\rho+\gamma)}\left(\frac{1}{\rho+\gamma} - \theta\right) + \frac{k\beta^2}{\gamma\mu(\rho+\gamma)}\left[\frac{1}{\rho+\gamma} - (1-\theta)\right] \right\}(1 - e^{-\gamma t}) > 0$$

$$Q^* - Q_2^* = \left[ \frac{k\alpha^2}{\gamma\lambda(\rho+\gamma)}\left(\frac{1}{\rho+\gamma} - \theta\right) + \frac{k\beta^2}{\gamma\mu(\rho+\gamma)}\left(\frac{1}{\rho+\gamma} - \frac{1+\theta}{2}\right) \right](1 - e^{-\gamma t}) > 0$$

$$Q^* - Q_3^* = \left\{ \frac{k\alpha^2}{\lambda\gamma(\rho+\gamma)}\left(\frac{1}{\rho+\gamma} - \frac{2-\theta}{2}\right) + \frac{k\beta^2}{\mu\gamma(\rho+\gamma)}\left[\frac{1}{\rho+\gamma} - (1-\theta)\right] \right\}(1 - e^{-\gamma t}) > 0$$

From the above two equations, Corollary 3 is obtained.

**Corollary 3.** *When the profit distribution $\theta \in (\frac{1}{3}, \frac{2}{3})$, under centralized decision making, PG is superior to PG under three scenarios of decentralized decision making.*

From Corollary 3, the main difference between the C2B model of the high-grade E-commerce model and the traditional E-commerce model is that consumers are more critical in the market, and they constantly collect the requirements of consumers for the product's appearance, quality, and functionality. Therefore, to improve the PG, the SC members need to form an efficient form of cooperation; the EP constantly collects feedback data from

consumers, forming the corresponding PG requirements for the manufacturer. According to the manufacturer's requirements for designing the product's appearance or investing in more research and development costs to improve product features to produce products that meet the needs of consumers. This exclusive partnership can significantly improve PG and increase consumer satisfaction with the product.

- Comparison of three scenarios under decentralized decision making

According to the function of the optimal value of the overall SC benefit under three different cases of decentralized decision-making, we can obtain

$$S_{c2}^* - S_{c1}^* = \frac{\beta^2 k^2 (3\theta - 1)^2}{8\mu\rho(\rho + \gamma)^2} > 0$$

$$S_{s2}^* - S_{s1}^* = \frac{\beta^2 k^2 (3\theta - 1)(1 - \theta)}{4\mu\rho(\rho + \gamma)^2} > 0$$

$$S_2^* - S_1^* = \frac{\beta^2 k^2 (3\theta - 1)(1 + \theta)}{8\mu\rho(\rho + \gamma)^2} > 0$$

$$S_{c3}^* - S_{c1}^* = \frac{\alpha^2 k^2 \theta(2 - 3\theta)}{4\lambda(\rho + \gamma)^2} > 0$$

$$S_{s3}^* - S_{s1}^* = \frac{\alpha^2 k^2 \theta(2 - 3\theta)^2}{8\lambda(\rho + \gamma)^2} > 0$$

$$S_3^* - S_1^* = \frac{\alpha^2 k^2 (\theta - 2)(3\theta - 2)}{8\lambda(\rho + \gamma)^2} > 0$$

The above six equations lead to Corollary 4.

**Corollary 4.** *When the profit distribution $\theta \in (\frac{1}{3}, \frac{2}{3})$, whether manufacturer-led or EP-led, the optimal profitability of manufacturers and EPs, as well as the SC as a whole, increases after manufacturers and EPs move from the Nash non-cooperative game to the Stackelberg master-slave game.*

From Corollary 4, it can be seen that if manufacturers and EPs simultaneously choose their own best QB level, the pursuit of their interests to maximize rather than choose to cooperate with SC members, for individual SC members and the overall benefits of the SC is very unfavorable. Thus, the need for manufacturers to share quality costs can achieve Pareto improvement of SC members to achieve a win-win situation for both manufacturers and EPs. Thus, even if the two sides are in Nash's non-cooperative game at the beginning, as cooperation deepens, they will transition into Stackelberg's master-slave game to improve their respective gains.

Based on the best QB of manufacturers and EPs in three different situations under decentralized decision-making, we can obtain

$$x_2^* - x_1^* = 0$$

$$y_2^* - y_1^* = \frac{\beta k(3\theta - 1)}{2\mu(\rho + \gamma)} > 0$$

$$x_3^* - x_1^* = \frac{\alpha k(2 - 3\theta)}{2\lambda(\rho + \gamma)} > 0$$

$$y_3^* - y_1^* = 0$$

The above four equations lead to Corollary 5.

**Corollary 5.** *When the profit distribution $\theta \in (\frac{1}{3}, \frac{2}{3})$, whether the manufacturer or EP dominates, the dominant party's best QB remains the same. The following party's best QB is enhanced in the Stackelberg master-slave game after the manufacturer and EP move from the Nash non-cooperative game to the Stackelberg master-slave game.*

According to the PG function for three different cases under decentralized decision-making, we obtain

$$Q_2^* - Q_1^* = \frac{k\beta^2(3\theta - 1)}{2\gamma\mu(\rho + \gamma)}(1 - e^{-\gamma t}) > 0$$

$$Q_3^* - Q_1^* = \frac{\alpha^2 k(2 - 3\theta)}{2\lambda\gamma(\rho + \gamma)}(1 - e^{-\gamma t}) > 0$$

Corollary 6 can be obtained from the above equation.

**Corollary 6.** *When the profit distribution $\theta \in (\frac{1}{3}, \frac{2}{3})$, whether PG has further enhanced manufacturer-led or EP-led, manufacturers and EPs after moving from the Nash non-cooperative game to the Stackelberg master-slave game.*

From Corollary 5 and Corollary 6, it is clear that when one party in the SC is the core member (i.e., the dominant party), its behavior largely determines the grade of the product. In order to gain more significant benefits from cooperation with other members of the SC, it will coordinate its QB so that its products meet the high-grade requirements in the C2B model. The QB cost of the follower is shared to promote successful cooperation with them. Once the cooperation is formed, the cost of QB of the follower is shared so that the QB can be better adjusted and improved, which makes the PG further improved and helps to increase the market share of the product, which can vastly improve the efficiency for both members of the SC.

In addition, the two Stackelberg master-slave games are compared according to the manufacturer-dominated and the EP-dominated. By calculation (the calculation process is omitted), when the profit distribution coefficient is $\theta \in (\frac{1}{3}, \frac{2}{3})$, there is no strict positive or negative difference between these two cases, depending on the size of each coefficient and the size of the distribution coefficient $\theta$. As to which game model the manufacturer and the EP will eventually evolve into in the actual game, it depends on who is in a stronger or weaker position. In order to facilitate the study, only one Stackelberg master-slave game is discussed below. The descriptions of Stackelberg master-slave games later in this paper are based on the manufacturer-dominated Stackelberg master-slave game.

*5.2. Coordination Mechanism Establishment*

From the above inference, it is clear that the highest level of QB and the highest PG are produced by the manufacturer and the EP under centralized decision-making. It also brings the highest overall SC profit and achieves Pareto optimality. However, centralized decision-making requires unified decision-making in the SC, which is practically impossible. This is because both manufacturers and EPs are rational and independent individuals, and it is not easy to make unified decisions through the SC.

Therefore, it is necessary to establish a cooperative coordination mechanism for both sides to still cooperate under the centralized decision-making situation to maximize the overall interests of the SC.

In the cooperative coordination model, the manufacturer and the EP will allocate the total SC profit under the centralized decision according to the ratio of and, so we obtain

$$S_{c4}^* = \frac{\theta}{\rho}\left(r + \frac{(k\alpha)^2}{2\lambda(\rho + \gamma)^2} + \frac{(k\beta)^2}{2\mu(\rho + \gamma)^2}\right) + \frac{k\theta}{\rho + \gamma}Q$$

$$S_{s4}^* = \frac{1-\theta}{\rho}\left(r + \frac{(k\alpha)^2}{2\lambda(\rho+\gamma)^2} + \frac{(k\beta)^2}{2\mu(\rho+\gamma)^2}\right) + \frac{k(1-\theta)}{\rho+\gamma}Q$$

According to principal–agent theory, the profits of both parties in the cooperative coordination model need to satisfy the individual rationality constraints of both the manufacturer and the EP. Then this model is Pareto optimal for both parties to be willing to cooperate and coordinate with each other. Therefore the following constraints need to be satisfied

$$S_{c4}^* - S_{c2}^* \geq 0 \tag{27}$$

$$S_{s4}^* - S_{s2}^* \geq 0 \tag{28}$$

Since the manufacturer, EP, and the overall profit of the SC in the Stackelberg master-slave game case are more significant than in the Nash non-cooperative game case, only the two constraints of Equations (27) and (28) must be satisfied to meet the requirements.

From Equation (27), we have

$$\frac{\theta k^2 \alpha^2 (1-\theta)}{2\lambda\rho(\rho+\gamma)^2} - \frac{k^2\beta^2(1-\theta)^2}{8\mu\rho(\rho+\gamma)^2} \geq 0 \tag{29}$$

From Equation (28), we have

$$\frac{(1-\theta)k^2\alpha^2(1-2\theta)}{2\lambda\rho(\rho+\gamma)^2} + \frac{k^2\beta^2(1-\theta)^2}{4\mu\rho(\rho+\gamma)^2} \geq 0 \tag{30}$$

From the concatenation of Equations (29) and (30), the range that satisfies the cooperative coordination model is

$$\frac{2\mu\alpha^2 + \lambda\beta^2}{4\mu\alpha^2 + \lambda\beta^2} \geq \theta \geq \frac{\lambda\beta^2}{4\mu\alpha^2 + \lambda\beta^2} \tag{31}$$

At the same time, $\theta$ should also satisfy the original interval $(\frac{1}{3}, 1)$. That is, the final range should take a part of the intersection of the range of Equation (31) and the interval to be the final $(\frac{1}{3}, 1)$ range that finally satisfies the cooperative coordination model.

## 6. Numerical Simulation and Sensitivity Analysis

### 6.1. Numerical Simulation

In order to visually express the results of the comparative analysis of optimal value, optimal QB, and optimal product quality under different game conditions, numerical simulations of the above differential model were performed. The specific assignment of each parameter is shown in Table 1.

**Table 1.** Parameter assignment.

| Parameter | $\alpha$ | $\beta$ | $\gamma$ | $\lambda$ | $\mu$ | $r$ | $k$ | $\rho$ | $Q_0$ |
|---|---|---|---|---|---|---|---|---|---|
| Assignment | 0.3 | 0.1 | 0.2 | 0.5 | 0.2 | 0.5 | 0.4 | 0.1 | 0.5 |

Where $\alpha$ and $\beta$ are the coefficients of the qualitative behavior of the manufacturer and the EP on PG, $\gamma$ is the decline rate of PG over time. $\lambda, \mu$ are the quality cost coefficients of the manufacturer and the EP. $r$ is the constant of the linear function of product SC revenue and PG. $k$ is the coefficient of the impact of PG on revenue. $\rho$ is the discount rate. $Q_0$ is the value of PG at the initial moment.

After assigning values according to specific parameters, the range of $\theta$ values in Equation (31) can be obtained as $\frac{5}{77} \leq \theta \leq \frac{41}{77}$. At the same time, $\theta$ should also satisfy the original interval $(\frac{1}{3}, 1)$; the final range of $\theta$ should be $\frac{1}{3} < \theta \leq \frac{41}{77}$. At this point, $\theta = \frac{1}{2}$ can be taken into the numerical simulation.

The equilibrium results of the Nash non-cooperative game under decentralized decision-making, the Stackelberg master-slave game dominated by the manufacturer, and the game under cooperative coordination mode can be obtained by calculation, as shown in Table 2.

**Table 2.** Comparison of equilibrium results in different game situations.

|  | Nash Non-Cooperative Game | Manufacturer-Driven Stackelberg Master-Slave Game | Cooperation and Coordination Model Game |
|---|---|---|---|
| $x^*$ | 0.40 | 0.40 | 0.80 |
| $y^*$ | 0.33 | 0.50 | 0.67 |
| $Q^*$ | $0.77 - 0.27e^{-0.2t}$ | $0.85 - 0.35e^{-0.2t}$ | $5.11 - 4.61e^{-0.2t}$ |
| $S_c^*$ | $0.67Q + 3.12$ | $0.67Q + 3.15$ | $0.67Q + 3.52$ |
| $S_s^*$ | $0.67Q + 3.41$ | $0.67Q + 3.47$ | $0.67Q + 3.52$ |
| $S^*$ | $1.34Q + 6.53$ | $1.34Q + 6.62$ | $1.34Q + 7.04$ |

From Table 2, we can directly obtain

$$x_1^* = x_2^* < x^* \tag{32}$$

$$y_1^* < y_2^* < y^* \tag{33}$$

This argues for the correctness of Corollary 2 and Corollary 5.

Matlab software is used to plot the images of the optimal revenue functions of manufacturers, EPs, and SCs as a function of time for different game situations, as shown in Figure 1.

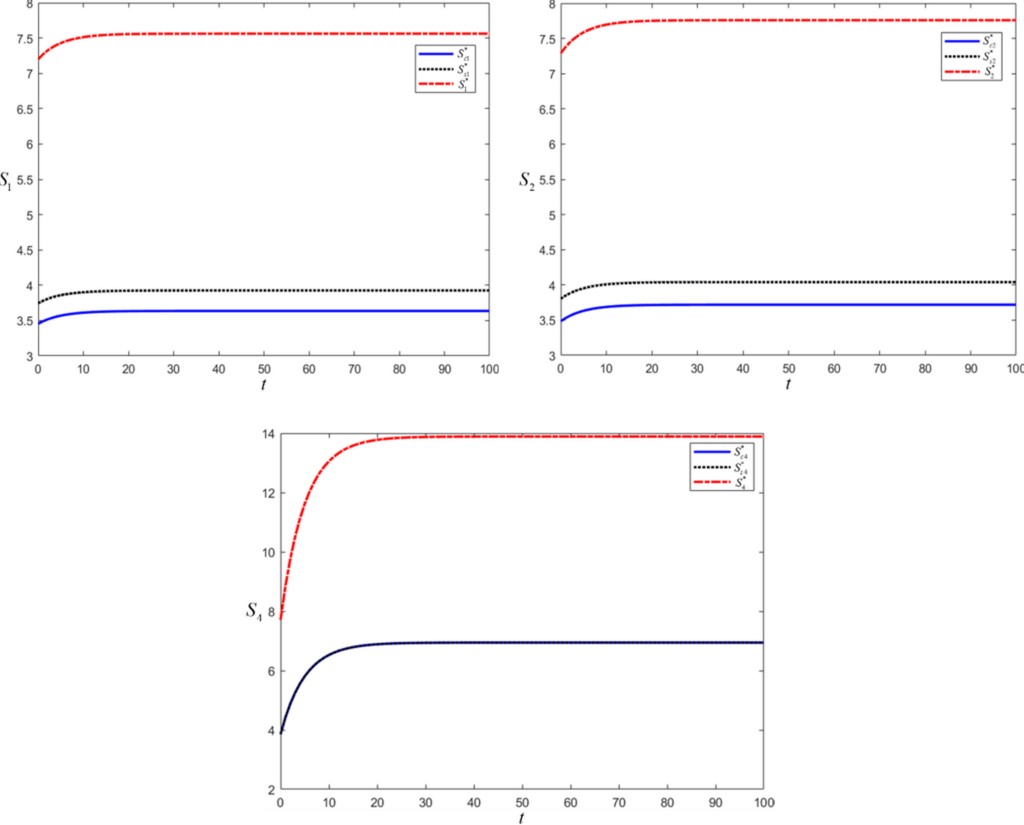

**Figure 1.** Optimal revenue function of manufacturer, EP and SC under three game scenarios.

The above figure shows that the optimal returns of manufacturers, EPs and SCs in the three-game scenarios have a time-stable tendency. Initially, the time-optimal revenue rises sharply with time, and then tends to level off gradually at the later stage, and finally reaches the equilibrium state.

The optimal PGs under the three models of the Nash non-cooperative game, Stackelberg master-slave game, and cooperative coordination model game are also plotted, and a visual comparison analysis is made through the images, as shown in Figure 2.

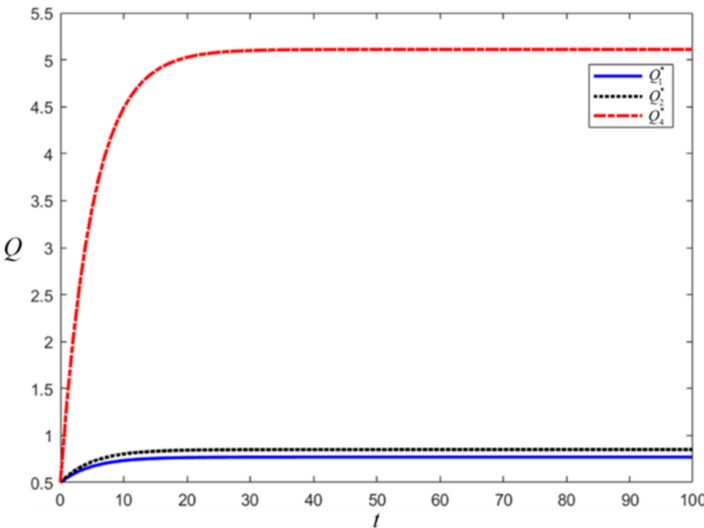

**Figure 2.** Comparison of optimal PG for three game scenarios.

From Figure 2, it can be visualized that PG under cooperative coordination mode is better than PG under the Stackelberg master-slave game, and PG under Stackelberg master-slave game is, in turn, better than PG under Nash non-cooperative game, which confirms the correctness of Corollary 3 and Corollary 6. Moreover, it can be obtained that the optimal PG, although increasing with time, is increasing more and more slowly, and the PG improvement shows a marginal decreasing trend, which is also consistent with reality.

Then draw the image of the optimal benefit function of the manufacturer, EP, and SC under the three-game situations and visually compare the level of their optimal benefits under the three games, as shown in Figure 3.

From the above figure, we can see that both manufacturers, EPs, and SCs have the most significant optimal benefit under the cooperative coordination model, followed by the Stackelberg master-slave game, confirming the correctness of Corollary 1 and Corollary 4. Therefore, the cooperative coordination model can effectively motivate both parties to participate and obtain benefits more significant than the maximum benefit under decentralized decision-making.

### 6.2. Sensitivity Analysis

In order to better understand the impact of changes in each vital parameter on the equilibrium solution of the DG under the cooperative coordination model, this subsection reveals the law of the magnitude of the impact of changes in essential parameters by using sensitivity analysis. Thus, we can better grasp the relevant laws and increase the total gain for both parties. Here, the effects of QB impact coefficients $\alpha$ and $\beta$, quality cost coefficients $\lambda$ and $\mu$, and PG impact coefficients $k$ on revenue on the DG equilibrium solution are studied. Moreover, refer to the parameter values assigned in Table 1 as the benchmark and the assumed time $t = 50$ before and after changing the parameter value size $\pm 10\%$ and $\pm 20\%$ to simulate the influence change.

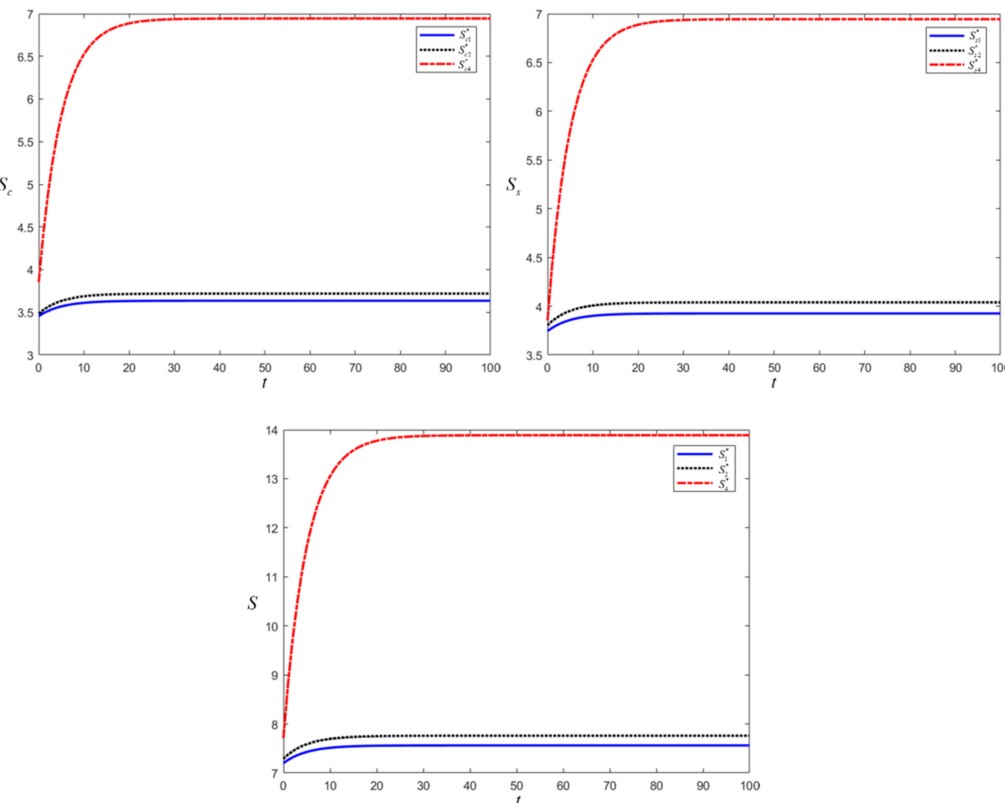

**Figure 3.** Optimal revenue function of manufacturer, EP and SC under three game scenarios.

- Effect of QB impact coefficient on equilibrium solution

The QB impact coefficients $\alpha$ and $\beta$ represent the coefficient of the degree of impact of QB on PG. It indicates the strength of the respective QB influence of the EP and the manufacturer, and the change of PG affects the overall SC revenue and PG. Therefore, it is necessary to study the influence of different sizes of QB influence coefficients $\alpha$ and $\beta$ on the equilibrium solution of the DG by keeping the other parameter values constant and changing the size of QB influence coefficients to observe the change in equilibrium solution size. Since there are two QB impact coefficients, they are analyzed in turn, as shown in Tables 3 and 4.

**Table 3.** Impact of QB coefficient $\alpha$ on equilibrium solution.

| Parameter Value | | $x^*$ | $y^*$ | $Q^*$ | $S_c^*$ | $S_s^*$ | $S^*$ |
|---|---|---|---|---|---|---|---|
| | 0.24 | 0.640 | | 3.671 | 5.682 | 5.682 | 11.363 |
| | 0.27 | 0.720 | | 4.351 | 6.271 | 6.271 | 12.542 |
| $\alpha$ | 0.3 | 0.800 | 0.667 | 5.111 | 6.930 | 6.930 | 13.859 |
| | 0.33 | 0.880 | | 5.951 | 7.658 | 7.658 | 15.315 |
| | 0.36 | 0.960 | | 6.871 | 8.455 | 8.455 | 16.910 |

**Table 4.** Impact of QB coefficient $\beta$ on equilibrium solution.

| Parameter Value | | $x^*$ | $y^*$ | $Q^*$ | $S_c^*$ | $S_s^*$ | $S^*$ |
|---|---|---|---|---|---|---|---|
| | 0.08 | | 0.533 | 4.711 | 6.583 | 6.583 | 13.166 |
| | 0.09 | | 0.600 | 4.900 | 6.747 | 6.747 | 13.493 |
| $\beta$ | 0.1 | 0.800 | 0.667 | 5.111 | 6.930 | 6.930 | 13.859 |
| | 0.11 | | 0.733 | 5.344 | 7.132 | 7.132 | 14.263 |
| | 0.12 | | 0.800 | 5.600 | 7.354 | 7.354 | 14.707 |

From Tables 3 and 4, it can be seen that as the QB impact factor $\alpha$ increases, the QB level of the EP remains constant, and the QB level of the manufacturer, PG, the respective interests of both parties, and the overall SC interests increase. As the QB impact factor $\beta$ increases, the QB level of the manufacturer remains the same, and the QB level of the EP, PG, the respective interests of both parties, and the overall interests of the SC all increase. The reason is that when the QB impact factor becomes more extensive, the influence of the party's QB on the PG increases, i.e., the level of the party's best QB increases, and the PG's QB increases because the factor becomes more extensive so that the party can reduce the cost of improving QB. Therefore, the overall SC benefits are increased, and the benefits of both parties are distributed in a fixed proportion to the overall SC benefits, and their profits will change in the same direction as the overall SC benefits.

- Effect of quality cost factor on equilibrium solution

The quality cost coefficients $\lambda$ and $\mu$ indicate the magnitude of the cost spent by the manufacturer and the EP to improve the level of QB. These two coefficients affect the cost of quality inputs to improve PG by both parties, which will affect the other values. The effect of $\lambda$ and $\mu$ on the equilibrium solution is analyzed by controlling the variables. The results are shown in Tables 5 and 6.

**Table 5.** Effect of quality cost factor $\lambda$ on equilibrium solution.

| Parameter Value | | $x^*$ | $y^*$ | $Q^*$ | $S_c{}^*$ | $S_s{}^*$ | $S^*$ |
|---|---|---|---|---|---|---|---|
| | 0.4 | 1.000 | | 6.111 | 7.796 | 7.796 | 15.592 |
| | 0.45 | 0.889 | | 5.555 | 7.315 | 7.315 | 14.629 |
| $\lambda$ | 0.5 | 0.800 | 0.667 | 5.111 | 6.930 | 6.930 | 13.859 |
| | 0.55 | 0.727 | | 4.747 | 6.614 | 6.614 | 13.228 |
| | 0.6 | 0.667 | | 4.444 | 6.352 | 6.352 | 12.703 |

**Table 6.** Effect of quality cost coefficient $\mu$ on equilibrium solution.

| Parameter Value | | $x^*$ | $y^*$ | $Q^*$ | $S_c{}^*$ | $S_s{}^*$ | $S^*$ |
|---|---|---|---|---|---|---|---|
| | 0.16 | | 0.833 | 5.389 | 7.171 | 7.171 | 14.341 |
| | 0.18 | | 0.741 | 5.234 | 7.036 | 7.036 | 14.072 |
| $\mu$ | 0.2 | 0.800 | 0.667 | 5.111 | 6.930 | 6.930 | 13.859 |
| | 0.22 | | 0.606 | 5.010 | 6.842 | 6.842 | 13.684 |
| | 0.24 | | 0.556 | 4.926 | 6.769 | 6.769 | 13.538 |

From Tables 5 and 6, it can be seen that as the quality cost factor $\lambda$ increases, the level of QB of the EP remains the same, and the level of QB of the manufacturer, PG, the respective interests of both parties, and the overall interests of the SC gradually decrease. As the quality cost factor $\mu$ increases, the QB level of the manufacturer remains the same. The QB level of the EP, PG, both parties' respective benefits, and the SC's overall benefits gradually decrease. The reason is that when the quality cost factor increases, the input cost of the company increases at the same level as QB. Therefore the party will reduce the quality cost input accordingly considering the cost factor, thus reducing the best QB level of the party. The PG becomes worse, and the increase in quality cost will also lead to the overall benefit of the SC, and the allocated benefit of both parties will also be reduced.

- Effect of PG on earnings coefficient on equilibrium solution

Nowadays, PG is one of the purchasing points that people are highly concerned about. People want to buy quality goods. Consumers may prefer high-priced high-grade goods to low-priced, low-grade goods. Therefore, PG impacts the commodity market, which will also be reflected in sales and revenue. Therefore, in the assumptions of the model, the coefficient $k$ of PG on revenue is set in the assumption of SC revenue. It is evident that $k$ affects the equilibrium solution of the DG between the interests of both parties. Therefore,

the value of *k* is varied to analyze the change in the equilibrium solution. The results are shown in Table 7.

**Table 7.** Effect of Coefficient *k* on equilibrium solution.

| Parameter Value | | $x^*$ | $y^*$ | $Q^*$ | $S_c{}^*$ | $S_s{}^*$ | $S^*$ |
|---|---|---|---|---|---|---|---|
| | 0.32 | 0.640 | 0.533 | 4.089 | 5.335 | 5.335 | 10.670 |
| | 0.36 | 0.720 | 0.600 | 4.600 | 6.088 | 6.088 | 12.176 |
| *k* | 0.4 | 0.800 | 0.667 | 5.111 | 6.930 | 6.930 | 13.859 |
| | 0.44 | 0.880 | 0.733 | 5.622 | 7.860 | 7.860 | 15.719 |
| | 0.48 | 0.960 | 0.800 | 6.133 | 8.879 | 8.879 | 17.757 |

As seen in Table 7, the level of QB of the manufacturer, the level of QB of the EP, PG, the respective interests of both parties, and the overall interests of the SC all increase as the coefficient *k* of the impact of PG on revenue continues to increase. This is because a larger *k* indicates that the market places more importance on PG. This will drive the participants to increase the importance of PG and the cost of quality to gain a specific position in the market. As a result, each participant's level of QB increases, and the PG becomes higher. The overall SC revenue increases as grades improve and *k* value increases. Thus, the profit of both parties goes in the same direction as the SC revenue.

## 7. Results Analysis, Discussions and Conclusions

In order to better improve the profits of both sides of the SSC and reasonably distribute the benefits, this paper focuses on the PG influence factors. It establishes a dynamically changing DG model with the SSC consisting of an E-commerce self-owned brand EP and a manufacturer as the research background. It also compares the equilibrium solutions of the DG under centralized decisions and decentralized decisions. The DG equilibrium solutions of the Nash non-cooperative game, manufacturer-dominated Stackelberg master-slave game, and EP-dominated Stackelberg master-slave game under decentralized decision-making are also compared. After a detailed comparative analysis, a reasonable, cooperative coordination mechanism is proposed to increase the overall benefits of manufacturers, EPs, and SCs to achieve Pareto optimality. At the same time, numerical simulations of the comparative analysis results are carried out by Matlab to visually verify the DG equilibrium solutions under different game situations. Finally, a sensitivity analysis is conducted to discuss the effects of three important parameters, namely, QB impact coefficients $\alpha$ and $\beta$, quality cost coefficients $\lambda$ and $\mu$, and PG impact coefficient *k* on revenue, on the DG equilibrium solution under the cooperative coordination mechanism.

### 7.1. Results Analysis

The above analysis leads to the following conclusions of this paper:

- Compared with the three cases under decentralized decision-making, the centralized decision-making scenario can obtain the maximum total SC profit, and the QB of SC members is improved.
- The optimal profit of the manufacturer and EP, as well as the SC as a whole, increases after switching from the Nash non-cooperative game to the Stackelberg master-slave game, whether the manufacturer or the EP dominates.
- After the manufacturer and EP shift from the Nash non-cooperative game to the Stackelberg master-slave game, the best QB of the dominant party remains unchanged, and the best QB of the following party is enhanced in the Stackelberg master-slave game, and the PG is further improved.
- The established cooperative coordination mechanism can ensure that the interests of each member are not less than the maximum interests of each party under decentralized decision-making and effectively motivate each member to participate in the mechanism.

- Under the sensitivity analysis of the cooperative coordination model, firstly, when the QB impacts factor $\alpha$ increases, the QB level of the party whose factors are multiplied remains the same, and the QB level of the party whose factors are not multiplied, PG, the respective interests of both parties, and the overall SC interests increase. Second, when the quality cost coefficient $\lambda$ increases, the QB of the party whose coefficients are multiplied remains the same, and the QB of the party whose coefficients are not multiplied, PG, the respective benefits of both parties, and the overall benefits of the SC decrease. Finally, when the coefficient $k$ of the impact of PG on revenue increases, the level of QB of the manufacturer, the level of QB of the EP, PG, the respective benefits of both parties, and the overall benefits of the SC all increase.

### 7.2. Discussions

The following insights were discussed from the conclusions of the model analysis:

- In the SSC network composed of an EP and its partner manufacturers, both parties take a centralized decision-making situation; SC members should take reasonable PG control measures to coordinate the QB among each other effectively. Both parties' QB will improve in the process of mutual cooperation, which can promote further improvement of PG and will have greater competitiveness in the high-grade E-commerce market. It can be seen that when both parties make centralized decisions, the SC members are no longer independent individuals, limited by their interests. Instead, they look at the whole SC, thus increasing the overall benefits of the SC. This also requires the managers of both companies not to focus on their gains and losses but to look at and solve problems more systematically to establish a long-term and stable cooperative relationship to achieve maximum benefits.
- In the SSC network composed of EPs and their cooperative manufacturers, manufacturers should pay more attention to the quality cost-sharing of products in enterprises' daily SC operation management. Then refer to the coordination mechanism established and combine the actual situation of both parties to make reasonable revenue distribution to stimulate the cooperation and mutual assistance between the manufacturer and the EP and ultimately achieve the Pareto improvement of both SC members. This way, there is an opportunity to significantly improve the PG, establish a good reputation and brand image through their product power, and accumulate more customer sources and then enhance their recognition of the C2B model of high-grade E-commerce. Ultimately, the benefits of SC members will be maximized.
- In the SSC network consisting of EPs and their partner manufacturers, the following recommendations are made to improve the overall profitability of the SC. First, both parties should improve the influence of their respective QBs in the SC. For example, the EP can adopt strict audit standards to ensure the quality of products produced by the partner manufacturer. Manufacturers can optimize and improve the quality inspection process and procedures in the assembly line production to develop more stringent product qualification standards; secondly, improving QB does not mean increasing quality cost investment. Both sides should choose the way of higher input and output. For example, manufacturers can summarize and analyze the experience of quality management of peer enterprises and apply the current advanced quality management theory to create a set of quality management systems applicable to their own; finally, manufacturers need to cooperate with EPs when producing grade-better products. Promote the high-grade E-commerceC2B model and publicize the advantages of PG. We strive to increase customers' attention to PG so that the product can stand out among competing products and be recognized and favored by the market.

### 7.3. Conclusions

This paper investigates an innovative DG model based on relevant literature and theoretical foundations to establish a coordination mechanism after a comparative analysis of various decision-making scenarios. We conduct numerical simulation and sensitivity

analysis to grasp the related laws. This study explores how to maximize the benefits of the supply chain under the C2B model and provides a significant direction for the future of quality E-commerce.

However, there are still some limitations to this study. Firstly, to simplify the model, this paper only considers the impact of PG factors on SC returns. However, in the quality E-commerce C2B model, many other factors can affect the revenue situation of these SC members, such as the manufacturer's production process. By analyzing the dynamics of this factor over time, it can improve and extend the DG model. Secondly, this paper suggests the appropriate assumption so that the DG is based on complete and accurate information, but does not consider the DG with asymmetric information and incomplete clarity of the game process. The research can be further extended and improve the richness of future studies.

**Author Contributions:** Conceptualization, B.X. and Z.Z.; methodology, Z.Z.; software, Z.Z.; validation, B.X., Z.Z. and X.L.; formal analysis, B.X.; investigation, B.X.; resources, B.X.; data curation, B.X.; writing—original draft preparation, Z.Z.; writing—review and editing, X.L.; visualization, X.L.; supervision, Z.Z. All authors have read and agreed to the published version of the manuscript.

**Funding:** This research received no external funding.

**Institutional Review Board Statement:** Not applicable.

**Informed Consent Statement:** Not applicable.

**Data Availability Statement:** The data comes from Web of Science.

**Conflicts of Interest:** The authors declare no conflict of interest.

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
