# Peer review of "Research on the Coordination of Quality Behavior of Supply 3 Chain of E-Commerce Platform under C2B Model of High-Grade E-Commerce Based on Differential Game"

_jtaer, doi:10.3390/jtaer17040071_

Round 1

Reviewer 1 Report

Research on the coordination of quality behavior of supply chain of e-commerce platform under C2B model of high-grade e-commerce based on differential game

this paper will study the coordination of supply chain interests by establishing a differential game model considering product grade factors in a secondary supply chain composed of an e-commerce platform and individual manufacturers. By solving the equilibrium solution of the differential game under the centralized and decentralized decisions and comparing them, we propose a cooperative coordination mechanism to achieve the optimal interests of the supply chain members based on this. At the same time, numerical simulation is used to visually verify and illustrate the differential game equilibrium solutions under different game situations. Finally, a sensitivity analysis of several vital parameters, such as the quality behavior influence coefficient, is conducted, and the effects of different model coefficients on the differential game equilibrium solution are discussed. It is found that (1) the supply chain parties have optimal quality behavior in the centralized decision-making situation, and the overall benefit is the greatest. (2) Compared with the Nash non-cooperative game, the optimal quality behavior of the dominant party remains unchanged, and the optimal quality behavior of the following party is enhanced after both parties move from the Stackelberg master-slave game, and the optimal profits of both parties as well as the overall increase. (3) The cooperative coordination model can coordinate the quality behavior of both sides of the supply chain when the benefit distribution coefficient is within a specific range. I would suggest you to shorten the abstract and only pinpoint the key information

l   The title needs to revise. coordination of quality behavior of supply 3 chain of e-commerce platform”

l    To be legible, the whole text must be completely edited with the help of a native English editor to polish your writing to prevent redundancies, grammatical errors and punctuation problems. For instance, “this paper will study the coordination of supply chain interests by establishing a differential game model considering product grade factors in a secondary supply chain composed of an e-commerce platform and individual manufacturers”. Future tense should not appear here”

l   Please underscore the scientific value added/contributions of your paper in your abstract and introduction and address your debate shortly in the abstract. What has been studied Introduction should be clearly stated research questions and targets first. Then answer several questions: Why is the topic important (or why do you study on it)? What are research questions or objectives? What are your contributions? Why is to propose this particular method (This must come from Literature discussion)?

l   For your references, Ivan Darma Wangsa, Iwan Vanany & Nurhadi Siswanto (2022) An optimization model for fresh-food electronic commerce supply chain with carbon emissions and food waste, Journal of Industrial and Production Engineering, DOI: 10.1080/21681015.2022.2099473; Chunyan Zhu, Xu Guo & Shaohui Zou (2022) Impact of information and communications technology alignment on supply chain performance in the Industry 4.0 era: mediation effect of supply chain integration, Journal of Industrial and Production Engineering, DOI: 10.1080/21681015.2022.2099472

l   The literature review is necessary for you to clarify the “contribution” of your study. Each reference mentioned should be discussed; otherwise, it is not helpful just to list them or in tables.

l   Please make sure your conclusions' section underscore the scientific value added of your paper, and/or the applicability of your findings/results, as indicated previously. Basically, you should enhance your findings, limitations, underscore the scientific value added of your paper, and/or the applicability of your contributions/shortages and future study in this session

Reviewer 2 Report

Thank you for the opportunity to review this article. After reading this study, I recommend some major revisions are required to improve the manuscript for higher quality, as follows:

1.       The authors must provide the full name of term “C2B” because some readers may not understand it firsthand.

2.       Rationale in the abstract is not sufficient for leading to the goal of the research.

3.       It is remiss that for the Introduction, references are totally missing. Thus, there’re supposed to be bias information and insufficient literature in the standard Introduction. Also, background of the paper is insufficient, need discussing more.

4.       “Figure” and “Fig” should be unified

5.       Section 7 is unacceptable in terms of its organization. Thus, I suggest revisiting this part. First, it’s too long to follow. Second, this part must be separated into: Results Analysis, Discussions and Conclusions, including the limits and directions of future research.

Round 2

Reviewer 1 Report

This manuscript is revised .